# Italian Validation of the Online Student Engagement Scale (OSE) in Higher Education

**DOI:** 10.3390/bs13040324

**Published:** 2023-04-10

**Authors:** Francesco Sulla, Rachel Harrad, Alice Tontodimamma, Pierpaolo Limone, Antonio Aquino

**Affiliations:** 1Department of Human Studies, University of Foggia, 71121 Foggia, Italy; 2School of Psychology, Faculty of Medicine, Health and Life Science, Swansea University, Swansea SA2 8PP, UK; 3Department of Economics, University of Studies ‘G. D’Annunzio’ Chieti and Pescara, 66100 Chieti, Italy; 4Department of Humanities, Pegaso Online University, 80143 Napoli, Italy; 5Department of Neuroscience, Imaging and Clinical Sciences, University of Studies ‘G. D’Annunzio’ Chieti and Pescara, 66100 Chieti, Italy

**Keywords:** online engagement, university students, online learning, higher education

## Abstract

During the COVID-19 pandemic, entire university courses were moved online. This represented a challenge for universities, who were required to move toward an entirely online learning environment without adequate time to manage the change from traditional courses to online courses. However, beyond the emergency of the pandemic, higher education does increasingly incorporate an online learning element, and such a provision does appear to reflect both the desires of modern-day students and university offerings. For this reason, assessing students’ online engagement is fundamental, not least because it has been seen to be related both to students’ satisfaction and their academic achievement. A validated measure of student online engagement does not exist in Italy. Therefore, this study aims to assess both the factor structure and the validity of the Online Student Engagement (OSE) Scale in the Italian context. A convenience sample of 299 undergraduate university students completed a series of online questionnaires. The Italian OSE scale presents good psychometric properties and represents a valuable instrument for both practitioners and researchers examining students’ engagement in online learning.

## 1. Introduction

For many decades, both researchers and academic institutions have invested much effort and attention into understanding students’ engagement with their studies, with Hu and Kuh [1], for example, suggesting that this may be the most important factor influencing both learning and personal development during the college years. Despite the importance of student engagement within the academic context, there is not yet consensus on the definition of the term. In his pioneering work, Astin [2] defines student engagement as the extent to which students participate in learning through “the investment of physical and psychological energy” (p. 519). Skinner and Belmont [3] describe engagement as the intensity and quality of behavioural and emotional involvement during learning activities, whilst Kuh [4,5,6] suggests that engagement can be defined as the amount of time and effort students invest into academic activities relating to learning outcomes. A common theme amongst these definitions is the requirement of time and investment of resources into academic related activities.

It is also possible to differentiate between different types of student engagement. Handelsman et al. [7] make distinctions between affective and behavioural components of engagement, identifying four dimensions: Skills Engagement (such as keeping up with required reading and putting forth effort); Emotional Engagement (aspects such as making the course personally interesting or applying it to one’s own life); Participation/Interaction Engagement (having fun in sessions, participating actively in small group discussions); and Performance Engagement (doing well on tests, obtaining a good grade).

The reasons for a continued focus on academic engagement include the link to a variety of positive educational outcomes [8] as well as its seemingly protective effect against the discontinuation of students’ studies. It is well-documented that the more engaged students are with their studies, the greater their desire is to persist with their academic career [9,10,11].

Traditionally, academic engagement has been considered in the context of ‘in person’ education; however, in recent years there has been an increase in the number of courses which involve education delivered online. Higher education increasingly incorporates an online learning element in student learning, and online learning does appear to reflect the desires of modern-day students and university offerings. There are various ways of defining online learning. For example, Moore and Kearsley [12], in their definition, focus on the geographical distance between the instructor and students and where the majority of the content is remotely engaged with. Meanwhile, Allen and Seaman [13] provide a more precise definition of online learning, stating that these are courses where at least 80% of the material is delivered online. Whilst elements of online learning have been incorporated over time into many higher education offerings, in response to the COVID-19 pandemic, education providers transferred into entirely online formats, often with little time to manage the changes.

In many countries, including Italy, some aspects of the higher education provisions that were introduced in response to the pandemic remain, resulting in blended learning formats where courses integrate both online and in-person aspects [14,15]. Italy also has a number of universities offering courses delivered entirely online (amounting to 11 out of 94 institutions) [16]. Further, Massive Open Online Courses (MOOCs) are widespread in Italy and have an increasing number of student enrolments. EduOpen, for example, was created by a consortium of Italian universities to enable the delivery of MOOCs and currently offers 380 courses attended by more than 125,000 students [17,18]. Such provision enables access to education for students who would not otherwise be able to enrol in traditional universities and is also said to improve retention rates and to permit a response to space constraints in traditional settings [19]. With widespread access to the Internet in modern life and the rapid growth of online education, it is imperative that higher education institutions seek to provide quality online learning experiences. Whilst there are many advantages to online learning, there is also the risk of low student engagement and a sense of isolation amongst students [20,21]. For these reasons, it is vital for institutions to have available to them a method of assessment to ascertain students’ levels of engagement with their online studies in order to inform teaching practice. One such measure is the Online Student Engagement Scale (OSE) [22], which characterises online engagement as a multifaceted construct of four dimensions using the affective-behavioural conceptualization of engagement outlined by Handelsman et al. [7]. The four dimensions of the OSE Scale are: Skills (e.g., taking good notes when carrying out reading), Emotional Engagement (e.g., finding ways to make the course interesting), Participation (e.g., participating actively in small group discussion forums) and Performance (e.g., obtaining a good grade). The OSE Scale [21] was initially developed following a review of existing measures of student engagement. From this review, it was identified that the Student Course Engagement Questionnaire (SCEQ) [7] offered the most appropriate measure to develop further, given that it assessed engagement across a variety of factors. A pool of 30 behaviours that would operationalise these factors was devised by five instructors following a focus group, with amendments made to items in the SCEQ to reflect online courses. These 30 items were then piloted amongst the students in an online course, with the results suggesting a high level of reliability (Cronbach’s alpha = 0.95). Finally, the OSE Scale was tested across a larger and more diverse group of students, where factor analysis identified the four anticipated factors of Skills, Emotional Engagement, Performance and Participation, formed of 19 items. When combined, these items reportedly had a strong level of reliability (Cronbach’s alpha = 0.95).

The OSE Scale is widely cited and there is some evidence of its use outside of the Western world, although to the best of our knowledge, there are no attempts to adapt or validate the scale in this context. A translated version of the OSE Scale is employed by Xinlei and Yiyang [23] who report a Cronbach alpha of 0.82 in their assessment of online learning engagement amongst nursing students.

The OSE is widely used in English speaking countries to good effect, ascertaining factors that may predict online engagement [24], including stress perception and personality traits [25]. In one study [25], the relationship between the aspects of the OSE Scale and personality traits and stress is examined during the COVID-19 emergency. It finds that the personality trait of extraversion predicts participation and performance whilst neuroticism predicts both emotional and skills engagement, as well as performance. Agreeableness predicts participation whilst openness to experience predicts emotional engagement. Meanwhile, conscientiousness predicts all factors of student online engagement, whilst stress perceived as a hindrance is seen to negatively predict performance.

Other relevant factors influencing student online engagement have also been considered using the OSE Scale. Bollinger and Halupa [26], for example, focus on the relationship between online student engagement and transactional distance, which is the “distance of understandings and perceptions, caused in part by the geographic distance, that has to be overcome by teachers, learners and educational organizations if effective, deliberate, planned learning is to occur” (p. 2). Their results suggest that transactional distance predicts student engagement, such that as transactional distance decreases, levels of student engagement increase, thus illustrating the utility of this scale for modern university settings to consider ways to support students’ online learning.

Others [27] have identified relationships between learning satisfaction and online engagement using the OSE Scale, for example, noting that greater levels of students’ perceived learning satisfaction is associated with an increased likelihood of online learning engagement, whilst years of experience working post-qualification is associated with decreased levels of learning engagement, illustrating the applicability of the use of this tool for evaluation of higher education practice.

Despite its great utility, to the authors’ knowledge, the OSE Scale has not been translated into or validated in the Italian language. In fact, no such measure exists in Italy, which means there is not a validated measure available to assess students’ levels of engagement with their online courses. This is especially important as cultural aspects are a vital consideration when evaluating students’ engagement. Research has demonstrated that academic engagement potentially varies as a result of the different educational processes and cultural traditions of the countries in which universities sit. In her cross-cultural analysis, Shcheglova [28] observes that the levels of university student engagement vary across countries. Drawing on the work of Hofstede [29], she notes how countries can be differentiated by their individualistic and collectivist cultures, and in terms of Power Distance Indexes, highlights that these factors can influence educational practices and subsequent student engagement. This demonstrates the importance of evaluating both student engagement and educational practices in the context of cultural factors and the need for context-relevant instruments for the investigation of students’ engagement in online education.

Finally, it is important to note the benefits of this specific tool for the assessment of online engagement. Dixson [22] highlights three primary benefits of this tool: to enable research into the design of courses, to enable feedback on student engagement in response to course design and to enable the evaluation of the effectiveness of teaching. There is a justifiable need to validate this measure for use in Italy; therefore, this study aims to assess both the factor structure and the validity of the OSE Scale in an Italian context. We anticipated that the Italian version will demonstrate the same good psychometric properties observed in the validations of the English version of the OSE Scale. To test the convergent validity of the Italian OSE Scale, we correlate its dimensions with the Utrecht Work Engagement Scale, Student version in its short form (UWES-S-9) [30], as it is, to the best of our knowledge, the only reliable measure of university students’ engagement that has an Italian adaptation and validation. The UWES-S includes three dimensions: Vigour, Dedication and Absorption. More specifically, Vigour considers mental resilience, high energy levels and persistence in the face of difficulties; Dedication refers to high levels of involvement in one’s work, which also incorporates positive states such as enthusiasm, pride, and inspiration; whilst Absorption is defined as a positive state of full immersion in one’s work. We expect positive correlations amongst the sub-dimensions of the Italian OSE Scale and the UWES-S-9, given that they share the common background of student engagement, particularly regarding the affective components of engagement. However, we also anticipate that these scales will be sufficiently distinct constructs (correlation index below 0.70) given the differences between online learning and traditional (offline) learning.

## 2. Materials and Methods

### 2.1. Participants and Procedure

This study was conducted according to the ethical principles defined by the Declaration of Helsinki. It involves a convenience sample of 310 undergraduate university students enrolled at the University of Modena and Reggio Emilia (Italy) (response rate 62.2%). Eleven students (1.1% of the respondents) declared that they attended the course mainly in person rather than online, and so were excluded from analysis. The retained sample comprised 299 participants. Of these participants, 289 were female, aged between 20 and 48 (*M* = 23.29; *S.D.* = 4.99). The participants were either 1st or 3rd year students enrolled in an Education bachelor’s degree and taking part in a blended Psychology course. Participants did not receive any reward for participation.

Between May and June 2022, students were invited to participate in the study via the course’s Moodle platform. Those that expressed interest in participation received information sheets about the voluntary nature of their participation, the study and their role within it and their right to withdraw from the study at any time. Participants provided informed written consent to proceed and independently completed a self-administered and structured online questionnaire, which took around 10 min to complete.

The first section of the questionnaire aimed to assess demographic characteristics (i.e., age, gender, type of course). Then, all participants completed the OSE Scale [22] translated into Italian, and the Italian translation of the Utrecht Work Engagement Scale, Student version, short form (UWES-S-9) [30]. On the completion of the survey, participants were thanked and debriefed.

### 2.2. Instruments

Online Student Engagement Scale (OSE) [21,22] Italian translation.

Respondents indicated their level of online engagement with their course by indicating the extent to which 19 statements were characteristic of their behaviours using a 5-point Likert scale, from 1 (“not at all characteristic of me”) to 5 (“very characteristic of me”). Four dimensions of engagement were assessed: Skills (e.g., “taking good notes over readings, PowerPoints, or video lectures”), Emotional Engagement (e.g., “really desiring to learn the material”), Participation (e.g., “participating actively in small-group discussion forums”) and Performance (e.g., “getting a good grade”).

Utrecht Work Engagement Scale, Student version, short form; Italian translation (UWES-S-9), [30].

Respondents indicated their levels of engagement with their studies in terms of Vigour (e.g., “When I get up in the morning, I feel like going to class”), Absorption (e.g., “I get carried away when I am studying”) and Dedication (e.g., “I am enthusiastic about my studies”), indicating the extent to which 9 statements applied to them on a 7-point Likert scale, from 0 (Never) to 6 (Always, Every day).

### 2.3. Translation Process

Permission for translation and validation was received from the author of the original OSE Scale via email. The procedure of translation included three steps. First, two native English speakers independently translated the items of the scale into Italian (forward translation). One translator additionally had a psychology background and was the assessor of the translation. Next, two bilingual researchers, blind to the original version of the scale, independently back-translated the scale into English. These two new English versions were translated into Italian by two independent psychology researchers with a certificated knowledge of the English language, and blind to the original version (backward translation).

### 2.4. Analytical Approach

A preliminary analysis of the Italian OSE Scale was carried out using IBM SPSS Statistics for Windows vers. 27 and involved the detection of multivariate outliers via the Mahalanobis distance test [31]. We also assessed means, standard deviations and normal distribution by examining the indices of skewness and kurtosis.

To test the factor structure of the OSE scale in the Italian language, a Confirmatory Factor Analysis (CFA) was carried out. This was performed using the statistical software R [32], specifically, the Lavaan package [33]. The Diagonally Weighted Least Squares (DWLS) estimator was used. This method was chosen since an estimator of Maximum Likelihood (ML) for non-continuous variables (such as Likert scales) is not currently supported in the Lavaan library. The number of model parameters was 100, while the number of observations was 297.

To evaluate the CFA models, the goodness of fit was estimated using the Root Mean Square Error of Approximation (RMSEA), the Standardized Root Mean Square Residual (SRMR), the Comparative fit index (CFI) and the Tucker-Lewis index (TLI).

RMSEA is an absolute-fit index, in that it assesses how far a hypothesized model is from a perfect model, where a lower RMSEA value is better. A value less than 0.05 is generally considered a good fit. For SRMR, a value less than 0.08 is generally considered to be a good fit. CFI and TLI are incremental-fit indices that compare the fit of a hypothesized model with that of a baseline model, that is, a model with the worst fit [34]. Higher CFI and TLI values are better: the fit is good for values greater than 0.9. In the CFA, we tested a 4-factor model, following the original structure of the scale: Skills, Emotional Engagement, Participation and Performance. The internal consistency of the Italian translation of the OSE Scale was evaluated by assessing the Composite Reliability (CR). CR can be regarded as “an indicator of the shared variance among the observed variables used as an indicator of a latent construct” and can be calculated for each construct [35] (p. 384). According to Saunders, Lewis and Thornhill [36], a value of 0.7 or greater indicates that the items within a scale can be regarded as measuring the same variable of interest. We additionally computed measures of Average Variance Extracted (AVE). AVE provides an indication of the amount of variance provided by a construct relative to the amount of variance resulting from measurement error. Values exceeding 0.50 are generally considered acceptable.

In the Italian validation of the UWES-S-9 [30], the three dimensions (i.e., Vigour, Absorption, Dedication) showed adequate fit indices: RMSEA: 0.08, CFI: 0.96, TLI: 0.95. The authors also correlated the residuals for items 8 (‘I am immersed in my study’) and 9 (‘I get carried away when I am studying’), and item 1 (‘When I’m doing my work as a student, I feel bursting with energy’) and 5 (‘When I get up in the morning, I feel like going to class’). All the factor loadings were high, ranging from 0.62 to 0.91. The Cronbach alpha for the three subscales were above the cut-off of 0.70 and showed good internal consistency: α = 0.82 (95% CI: 0.79–0.85) for Vigour; α = 0.88 (95% CI: 0.86–0.90) for Dedication; and α = 0.76, (95% CI: 0.72–0.79) for Absorption. In this validation, the scale showed the invariance of the structure between Italian males and females, as well as between students of different ages and years of study (i.e., Bachelor’s years compared to Master’s years) [30].

## 3. Results

### 3.1. Preliminary Analyses and Descriptive Statistics

The Mahalanobis distance test indicated that two participants could be considered multivariate outliers with a distance value of 34.66 (*p* < 0.001) and 26.99 (*p* < 0.001), respectively. For this reason, we deleted these two participants from the analyses. The retained sample included 297 participants (287 females, *M* age = 23.29, *SD* = 5.02).

Table 1 reports the descriptive statistics (Mean, Standard Deviation, Skewness and Kurtosis) for the items of the Italian OSE Scale. Inspection of skewness and kurtosis values indicated that items are normally distributed (the indices are between −0.96 and 0.68), so no variable transformations are necessary.

### 3.2. Factor Structure of the Italian Translation of the OSE Scale

The solution of CFA converged in 29 iterations, yielding a model statistic of 588.037 with 146 degrees of freedom. CFA demonstrates that a 4-factor solution is a good fit for the data: RMSEA is 0.10, SRMR is 0.08, and CFI and TLI values are 0.97 and 0.96. In Table 2, fit indices for User versus Baseline model are reported.

Examining the factor loadings, all items have loadings greater than 0.30 on the expected factor (Table 3). Specifically, the six items of the Skills factor show factor loadings between 0.64 and 0.79, the five items of the Emotional factor show factor loadings between 0.75 and 0.83 and the six items of Participation factor show factor loadings between 0.65 and 0.82. Finally, the two items of Performance factor show factor loadings of 0.69 and 0.82.

In short, the CFA indicates that the Italian version of the OSE Scale shows a structure comparable to the original, with excellent fit indices and factor loadings on the expected factor.

The latent variables are positively correlated with each other (Table 4). The strongest correlation is observed between the Skills and Emotional engagement factors (Estimate = 0.65, *p* < 0.001, whereas a weaker correlation is observed between the Participation and Emotional engagement factors (Estimate = 0.46; *p* < 0.001).

The CR for the four dimensions of the Italian translation of the OSE Scale (Skills, Emotional Engagement, Participation and Performance) are 0.86, 0.89, 0.88, 0.72, respectively, thus confirming good reliability. In addition, the four dimensions of the translated OSE Scale show satisfactory values of AVE: Skills: 0.53, Emotional Engagement: 0.62, Participation: 0.55 and Performance: 0.57.

### 3.3. Convergent Validity

Before examining the correlations between the factors of the translated OSE Scale and the dimensions of the Italian UWES-S-9, we checked the reliability of this last scale in our sample. The three dimensions of the UWES-S-9 show good indices of CR and AVE: For Vigour, these values are 0.85 and 0.66, respectively, 0.87 and 0.69 for Dedication and 0.79 and 0.55 for Absorption.

For convergent validity, as expected, the factors of the Italian translation of the OSE scale positively correlate with the dimensions of the Italian UWES-S-9, supporting the suggestion that the OSE Scale is an appropriate tool in evaluating students’ engagement (Table 5). Using Cohen’s interpretation of r-values, we observe that Skills correlate highly with Vigour (r = 0.47; *p* < 0.001) and Absorption (r = 0.42; *p* < 0.001), and moderately with Dedication (r = 0.29; *p* <.001). Emotional Engagement correlates highly with Absorption (r = 0.42; *p* < 0.001) and Dedication (r = 0.45; *p* < 0.001), and moderately with Absorption (r = 0.34; *p* < 0.001). Participation correlates highly with Vigour (r = 0.47; *p* < 0.001) and moderately with Absorption (r = 0.31; *p* < 0.001) and Dedication (r = 0.24; *p* < 0.001), whilst Performance correlates moderately with the three dimensions of the UWES: Vigour (r = 0.33; *p* < 0.001), Absorption (r = 0.30; *p* < 0.001) and Dedication (r = 0.31; *p* < 0.001).

## 4. Discussion

The aim of this study is to provide an Italian version of the OSE Scale and to test its structure and validity. The results confirm the reliability and validity of the Italian version of the OSE Scale [21,22] on a Western European university student population from Italy. The confirmatory factor analysis suggests the same four-factor solution for the Italian version: Skills (e.g., taking good notes), Emotional Engagement (e.g., finding ways to make the course interesting), Participation (e.g., participating actively in small group discussion forums) and Performance (e.g., obtaining a good grade). The four dimensions show good reliability and good internal consistency. Finally, as expected, the factors of the OSE Scale positively correlate with the dimensions of the UWES-S [30], demonstrating the appropriateness of these scales for evaluating student engagement, which as online learning increases, becomes more important.

Numerous studies have linked student engagement with a variety of positive educational outcomes, [8] such as academic achievement. Extensive empirical research on the relationship between student engagement and academic achievement exists, and whilst results are not always consistent, e.g., [37,38], a recent meta-analysis [39] analysing 69 independent samples found a moderately strong and positive correlation between overall student engagement and academic achievement. As such, measuring students’ engagement in a reliable way is vital. Blended or online-only courses provide instructors unique opportunities to monitor engagement by using the trace data collected by the learning environment. However, most higher education instructors are not educational researchers or data specialists. Thus, the challenges of accessing and using some types of data might be a barrier to successfully monitoring student engagement [40]. As such, there is a benefit to instructors in employing the OSE Scale. Whilst achievement is not the only goal of an education system, it is a common measure of its success. It is used to evaluate, together with other indices, both the performance of schools and universities and to measure changes in individual students’ level of achievement.

Student engagement has also been seen to have an influence on students’ satisfaction. For example, Rajabalee and Santally [41] find a significant positive relationship between satisfaction and engagement (measured with the OSE Scale) in a sample of 844 first year university students across disciplines, as well as a weak but positive significant correlation between satisfaction and engagement with students’ overall performance. Moreover, Baloran et al. [42] find that satisfaction with an online course is significantly correlated with online student engagement in a sample of 529 university Filipino students during the COVID-19 pandemic. Through structural equation modelling, it is further demonstrated that online course satisfaction is significantly related to students’ skills engagement, emotional engagement, participation and performance, demonstrating the value of assessment of online engagement as part of a range of considerations.

## 5. Limitations and Conclusions

The study has some limitations that need to be considered. Firstly, it is important to note that our results are based on a single convenience sample and are limited by the sociocultural characteristics of the university settings where the study was conducted. Secondly, the percentage of females in the sample is high (96%) and this does not allow us to verify any gender differences in the levels of our variables. Thirdly, engagement is measured amongst students enrolled in a single course, and this may prevent generalisation to students enrolled in other majors and the university student population in general. The presence of only two items assessing the performance dimension is another limitation; however, this limitation follows the original version of the scale [21,22]. Another limitation regards the use of self-report measures and the possibility that this format may be impacted by socially desirable responses and measurement bias [43].

Despite these limitations, this work provides a useful tool for Italian researchers and academics to assess students’ online engagement. Assessing student engagement in online learning is fundamental in analysing students’ learning processes, and the level of engagement has the potential to be an indicator of online learning effectiveness. Overall, we provide evidence for the good psychometric properties of the Italian translated OSE Scale and suggest that it is a useful instrument for researchers and practitioners in several domains.

Future studies should now investigate university students’ online engagement in a more diverse sample to verify any potential differences. This Italian version of the OSE Scale will enable cross-cultural comparisons as well as comparative studies in general. The evidence of cultural differences in students’ engagement [28] acts to further justify the validation of this measure in the Italian language. Hofstede [29] previously reported a medium Power Distance Index amongst the Italian population. Studies using the OSE Scale can now assess this in a population of university students and compare students from different parts of the country (i.e., North vs. South vs. Isles). Future research is now able to consider Italian students’ online engagement as well as other variables linked to both their learning satisfaction and academic achievement to improve student experience. In short, this Italian OSE scale can be used to investigate, evaluate and further improve the higher educational landscape.

## Figures and Tables

**Table 1 behavsci-13-00324-t001:** Descriptive statistics of the Italian OSE Scale items.

Items	Mean	SD	Skewness	Kurtosis
S1: Assicurarsi di studiare con regolarità (*Making sure to study on a regular basis*)	3.61	0.91	−0.47	0.14
S2: Stare al passo con le lezioni (*Staying up on the readings*)	3.55	1.11	−0.38	−0.63
S3: Controllare gli appunti prima della lezione per essere certo di comprendere il materiale del corso (*Looking over class notes*)	3.07	1.12	−0.11	−0.74
S4: Essere organizzatə (*Being organized*)	3.99	0.99	−0.84	0.28
S5: Prendere degli appunti accurati sulle lezioni, in PowerPoint o le video lezioni (*Taking good notes over readings, PowerPoints or, video lectures*)	3.81	1.06	−0.71	−0.12
S6: Ascoltare/leggere attentamente (*Listening/reading carefully*)	4.08	0.78	−0.68	0.68
E1: Metterci molto impegno (*Putting forth effort*)	4.18	0.76	−0.76	0.67
E2: Trovare modi per rendere il materiale del corso pertinente con la mia vita (*Finding ways to make the course materials relevant to my life*)	3.84	0.96	−0.49	−0.27
E3: Applicare i contenuti del corso alla mia vita (*Applying course material to my life*)	3.95	0.91	−0.61	0.01
E4: Trovare modi per rendere il corso interessante per me (*Finding ways to make the course interesting to me*)	3.84	0.83	−0.33	−0.27
E5: Avere davvero voglia di imparare il materiale del corso (*Really desiring to to learn the material*)	4.15	0.77	−0.62	0.19
PA1: Divertirsi nelle chat online, discussioni o via email con il docente o gli altri studenti (*Having fun in online chats*)	2.75	1.12	0.15	−0.72
PA2: Partecipare in maniera attiva in discussioni in piccolo gruppo su forum (ad es. whatsapp, etc.) (*Participating actively in small-group discussion forums*)	3.03	1.08	0.04	−0.57
PA3: Aiutare ɜ colleghɜ (*Helping fellow students*)	3.91	0.87	−0.62	0.17
PA4: Partecipare a conversazioni online (chat, discussioni, email) sul corso (*Engage in conversations online chat, discussions, email*)	2.99	1.09	0.07	−0.58
PA5: Postare sul forum della piattaforma online (es. dolly, Teams) con regolarità (*Posting in the discussion forum regularly*)	2.57	1.23	0.37	−0.87
PA6: Riuscire a conoscere altr * collegh * che frequentano il corso (*Getting to know other students in the class*)	3.47	1.12	−0.45	−0.41
PER1: Voler ottenere un buon voto (*Getting a good grade*)	4.43	0.69	−0.96	0.13
PER2: Avere buoni risultati nei [in eventuali] test/quiz [in itinere] (es. Kahoot) (*Doing well in the tests/quizzes*)	3.91	0.97	−0.71	0.17

**Table 2 behavsci-13-00324-t002:** Fit indices of User Model (Standard) versus Baseline (Standard) Model.

Fit Index	Standard	Robust
CFI	0.97	0.89
TLI	0.96	0.88
RMSEA	0.10	0.11
RMSEA 90% Confidence Interval-Lower	0.09	0.10
RMSEA 90% Confidence Interval-Upper	0.11	0.12
*p* Value RMSEA	<0.001	<0.001
SRMR	0.08	0.08

**Table 3 behavsci-13-00324-t003:** Results of CFA: Estimates (EST), Standard Error (SE), z-value and significance (*p*) values are reported for each item of the OSE scale (the original item is reported in brackets).

	EST	SE	z-Value	*p*
Skills =~				
S1: Assicurarsi di studiare con regolarità (*Making sure to study on a regular basis*)	0.74	0.03	23.86	<0.001
S2: Stare al passo con le lezioni (*Staying up on the readings*)	0.64	0.03	17.04	<0.001
S3: Controllare gli appunti prima della lezione per essere certo di comprendere il materiale del corso (*Looking over class notes…*)	0.74	0.03	24.73	<0.001
S4: Essere organizzatǝ (*Being organized*)	0.69	0.04	18.98	<0.001
S5: Prendere degli appunti accurati sulle lezioni, in PowerPoint o le video lezioni (*Taking good notes over readings, PowerPoints or, video lectures*)	0.75	0.03	25.63	<0.001
S6: Ascoltare/leggere attentamente (*Listening/reading carefully*)	0.79	0.03	26.06	<0.001
**Emotional =~**				
E1: Metterci molto impegno (*Putting forth effort)*	0.75	0.04	18.79	<0.001
E2: Trovare modi per rendere il materiale del corso pertinente con la mia vita (*Finding ways to make the course materials relevant to my life*)	0.83	0.03	31.97	<0.001
E3: Applicare i contenuti del corso alla mia vita (*Applying course material to my life*)	0.78	0.03	28.54	<0.001
E4: Trovare modi per rendere il corso interessante per me (*Finding ways to make the course interesting to me*)	0.83	0.03	31.28	<0.001
E5: Avere davvero voglia di imparare il materiale del corso (*Really desiring to to learn the material*)	0.75	0.03	24.50	<0.001
**Participation =~**				
PA1: Divertirsi nelle chat online, discussioni o via email con il docente o gli altri studenti (*Having fun in online chats…*)	0.76	0.03	27.38	<0.001
PA2: Partecipare in maniera attiva in discussioni in piccolo gruppo su forum (ad es. whatsapp, etc.) (*Participating actively in small-group discussion forums*)	0.79	0.03	30.54	<0.001
PA3: Aiutare ɜ colleghɜ (*Helping fellow students*)	0.65	0.05	14.35	<0.001
PA4: Partecipare a conversazioni online (chat, discussioni, email) sul corso (*Engaging in conversations online (chat, discussions, email)*)	0.82	0.02	33.82	<0.001
PA5: Postare sul forum della piattaforma online (es. dolly, Teams) con regolarità (*Posting in the discussion forum regularly*)	0.71	0.03	22.01	<0.001
PA6: Riuscire a conoscere altrɜ colleghɜ che frequentano il corso (*Getting to know other students in the class*)	0.69	0.04	18.29	<0.001
**Performance =~**				
PER1: Voler ottenere un buon voto (*Getting a good grade*)	0.69	0.05	13.97	<0.001
PER2: Avere buoni risultati nei (IN EVENTUALI) test/quiz (IN ITINERE) (es. Kahoot) (*Doing well in the tests/quizzes*)	0.82	0.05	16.14	<0.001

**Table 4 behavsci-13-00324-t004:** Zero-order correlations among the latent variables (Oblimin Rotation): Estimates (EST), Standard Error (SE), z-value and significance (*p*) values.

	EST	SE	z-Value	*p*
**Skills ~~**				
Emotional	0.65	0.03	18.51	<0.001
Participation	0.59	0.04	13.47	<0.001
Performance	0.59	0.05	10.87	<0.001
**Emotional ~~**				
Participation	0.47	0.05	10.23	<0.001
Performance	0.46	0.06	7.38	<0.001
**Participation ~~**				
Performance	0.59	0.06	10.41	<0.001

**Table 5 behavsci-13-00324-t005:** Zero-order Correlations between the Italian OSE Scale and the Italian UWES-S-9 dimensions.

	Vigor	Absorption	Dedication
Skills	0.47 ***	0.42 ***	0.29 ***
Emotional	0.34 ***	0.42 ***	0.45 ***
Participation	0.47 ***	0.31 ***	0.24 ***

*** *p* < 0.001.

## Data Availability

The data presented in this study are available on request from the corresponding author. The data are not publicly available due to privacy reasons.

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
