# Peer review of "Italian Validation of the Online Student Engagement Scale (OSE) in Higher Education"

_behavsci, 2023, doi:10.3390/bs13040324_

Round 1

Reviewer 1 Report

This reviewer found the submitted manuscript to be interesting and engaging. It was very well written, appropriately structured, and presented the purpose, methodology, and results in a particularly clear manner. The work, if published, will most likely be a valuable addition to scholarly knowledge in this area, an area that has gathered increasing importance over the last few years as educational institutions have increasing moved, or have been compelled to move, in the direction of online and hybrid course presentation.

A significant issue that this study raises is the impact of national culture on student engagement and of its measurement by instruments such as the OSE scale. This does not appear to be addressed in the current version of the manuscript. Was there an initial sense that measurement of student engagement would be impacted, or moderated, by the national culture of the students involved? Was there a sense that Italian students might perform differently than, say, US students or those of other European countries? It might be helpful if the authors considered addressing this issue (e.g Shcheglova, I. (2018). A cross-cultural comparison of the academic engagement of students. Russian Education & Society, 60:  665-681. 10.1080/10609393.2018.1598163).

Author Response

This reviewer found the submitted manuscript to be interesting and engaging. It was very well written, appropriately structured, and presented the purpose, methodology, and results in a particularly clear manner. The work, if published, will most likely be a valuable addition to scholarly knowledge in this area, an area that has gathered increasing importance over the last few years as educational institutions have increasing moved, or have been compelled to move, in the direction of online and hybrid course presentation.

A significant issue that this study raises is the impact of national culture on student engagement and of its measurement by instruments such as the OSE scale. This does not appear to be addressed in the current version of the manuscript. Was there an initial sense that measurement of student engagement would be impacted, or moderated, by the national culture of the students involved? Was there a sense that Italian students might perform differently than, say, US students or those of other European countries? It might be helpful if the authors considered addressing this issue (e.g Shcheglova, I. (2018). A cross-cultural comparison of the academic engagement of students. Russian Education & Society, 60:  665-681. 10.1080/10609393.2018.1598163).

R: We want to thank reviewer 1 for the positive feedback. We appreciated their consideration and time in reviewing our manuscript. Also, we really want to thank the reviewer for raising this point, and, in particular, for suggesting the paper by Shcheglova (2018). It has provided much food for thought. We have now addressed this issue both in the introduction (i.e., lines 201-211) and discussion section (i.e., 1409-1415). Thanks again for your valuable comments.

Reviewer 2 Report

Dear Editor,

Thank you for allowing me to review this manuscript. The authors utilizing a cross-sectional design study with a convenience sample of 299 University undergraduate Italian students assessed both the factor structure and the psychometric properties of the Online Student Engagement (OSE) Scale. I suggest that this work should be published but before that, some points should be fixed.

The authors should present the previous efforts for the establishment of the factorial structure of the OSE.

The authors should provide permissions and ethical approvals for the conducted study. Moreover, a response rate is required.

I think that the authors had to follow a data screening approach before analyzing the data.  Maybe, some multivariate outliers are found with Mahalanobis' distance. These participants should be excluded from the analysis.

A descriptive statistics table (with mean, SD, skewness, and kurtosis) for each item, is required. It will help us to understand the items’ distribution.

Pay attention to line 151. "Eleven students (1.1% of the respondents)". It is not clear maybe there's a fault.

The authors should explain why did they use this estimator and not the ML? With references, please. Moreover, the authors should mention the corresponding package of R for this analysis (with reference).

I suggest to the authors calculate a more robust alternative index for reliability (composite reliability). Moreover, the authors should calculate and present the reliability indices for the UWES-9 instrument. 

Line 201, pay attention to the reference type.

Finally, I suggest to the authors, a limitation is the fact that the study that uses self-reports runs the risk of socially desired responses and measurement bias.

Lavidas, K.; Papadakis, S.; Manesis, D.; Grigoriadou, A.S.; Gialamas, V. The Effects of Social Desirability on Students’ Self-Reports in Two Social Contexts: Lectures vs. Lectures and Lab Classes. Information 2022, 13, 491. https://doi.org/10.3390/info13100491

Author Response

Dear Editor,

Thank you for allowing me to review this manuscript. The authors utilizing a cross-sectional design study with a convenience sample of 299 University undergraduate Italian students assessed both the factor structure and the psychometric properties of the Online Student Engagement (OSE) Scale. I suggest that this work should be published but before that, some points should be fixed.

The authors should present the previous efforts for the establishment of the factorial structure of the OSE.

R: We thank the reviewer for their positive feedback about the manuscript and for all comments that were important in improving the manuscript. As requested, we have now given more details about previous efforts to establish of the factorial structure of the OSE Scale (i.e., lines 103-113).

The authors should provide permissions and ethical approvals for the conducted study.

R: The study was conducted in accordance with the Declaration of Helsinki, following the general research principles and the ethical rules of the Italian Psychological Association (AIP). Informed consent was obtained from all subjects involved in the study (i.e., lines 595-596; 605-609).

Moreover, a response rate is required.

R: We now report a response rate (i.e., lines 597-598)

I think that the authors had to follow a data screening approach before analyzing the data.  Maybe, some multivariate outliers are found with Mahalanobis' distance. These participants should be excluded from the analysis

R: Following your suggestion, we have run Mahalanobis’s distance looking for multivariate outliers (i.e., lines 758-759). This analysis showed that two participants could be considered outliers (i.e., lines 978-981). Consequently, we have run again all analyses without them. The results were similar to the previous ones. We believe that the analyses without multivariate outliers are more reliable, for this reason we thank again the reviewer for this comment.

A descriptive statistics table (with mean, SD, skewness, and kurtosis) for each item, is required. It will help us to understand the items’ distribution.

R: We have added a descriptive statistics table in the new section entitled “Preliminary analyses and descriptive statistics” (i.e., lines 986-1073). It is worth noting that all items were normally distributed.

Pay attention to line 151. "Eleven students (1.1% of the respondents)". It is not clear maybe there's a fault. 

R: We apologize for not being clear on this point. The original sample included 310 participants. However, we discarded 11 participants that declared they attended the course mainly in person rather than online. We have now added this information in the Participants section (i.e., lines 596-603).

The authors should explain why did they use this estimator and not the ML? With references, please. Moreover, the authors should mention the corresponding package of R for this analysis (with reference).

R: Thank you for your comments that helped us to add missing information. To test the factor structure of the OSE scale in the Italian language a Confirmatory Factor Analysis (CFA) was performed using the statistical software R (R Core Team, 2022), specifically the Lavaan package (Rosseel, 2012). The Diagonally Weighted Least Squares (DWLS) estimator was used. This method was chosen since an estimator of Maximum Likelihood (ML) for non-continuous variables (such as Likert scales) is not currently supported in the Lavaan library. We have added this information in the manuscript (i.e., lines 762-767).

I suggest to the authors calculate a more robust alternative index for reliability (composite reliability). Moreover, the authors should calculate and present the reliability indices for the UWES-9 instrument

R: As requested, we have computed composite reliability and average variance extracted for OSE scale. We have added description or these indices in Analytic Approach Section (i.e., lines 778-786) and results (i.e., lines 1138-1207). Further, we calculate and present these indices also for UWES-9 instrument (i.e., lines, 956-975).

Line 201, pay attention to the reference type.

R: Thanks for pointing this out. The reference has now been deleted as we changed the analytic approach according to your comments.

Finally, I suggest to the authors, a limitation is the fact that the study that uses self-reports runs the risk of socially desired responses and measurement bias.

Lavidas, K.; Papadakis, S.; Manesis, D.; Grigoriadou, A.S.; Gialamas, V. The Effects of Social Desirability on Students’ Self-Reports in Two Social Contexts: Lectures vs. Lectures and Lab Classes. Information 2022, 13, 491. https://doi.org/10.3390/info13100491

R: We do agree with the reviewer and addressed this issue in the limitations section (i.e., lines 1398-1400)

Reviewer 3 Report

The issue of the article the validation of Online Student Engagement Scale in Italy. Authors have used confirmatory factor analysis and they analysed the relationship between the dimensions of UWES and OSE. The description of the translation process is detailed. In my opinion the empirical part is well-structured and well-prepared but the authors should improve some elements of the introduction and limitations.

I suggested some elements to be included in the theoretical chapter:

-        Information about the validation of OSE outside the Western world – are there any attempts? With what types of results?

-        Why can the Italian situation specific from the aspect of engagement? The description of it is a little bit superficial for me. Similar to the demonstration of the circumstances, possibilities and acceptance of online teaching.

-        I think the explanation of the selection of UWES can be very useful – why this scale was chosen? Authors should add some information about this scale in the introduction too.

-        The earlier empirical results of OSE scale can be more detailed.

-        If the UWES was used in Italy earlier, authors should describe these results. I can see in the 25th element of the reference list that this is a validated method in Italy but we have to know other details and results (if there are).

-        Is there any example for the co-usage of these too scale? Why did the authors choose?

I can see a significant limitation due to the sample and sample method. The engagement during the online courses can be different in the given disciplines and this sample belongs to only two institutions (and we can not see enough detailed the features of these institutions) and they come from only one discipline (Education) and one type of course (Psychology). The features of engagement, the working methods of the teaching can be disparate in the field of the other disciplines – e.g. in Science or Medicine.  I think the authors should empathize this part of the limitation more. The arte of women are very high too (289 from 299).

Perhaps the description of the sample can be more detailed (part or full time students), the number of the two subsamples (Reggio Emilia and University of Modena) and the patterns of the other socio-demographic variables (if they are given).

Future plans and the brief presentation of later research are recommended at the end of the paper (if the authors have this type of planes).

typos:

row 69 – Eduopean

row 244 – However (in the middle of the sentence)

row 59 – Universities (in the middle of the sentence)

Author Response

The issue of the article the validation of Online Student Engagement Scale in Italy. Authors have used confirmatory factor analysis and they analysed the relationship between the dimensions of UWES and OSE. The description of the translation process is detailed. In my opinion the empirical part is well-structured and well-prepared but the authors should improve some elements of the introduction and limitations.

Thank you for your supportive words about our work. In red, our replies to your comments.

I suggested some elements to be included in the theoretical chapter:

-        Information about the validation of OSE outside the Western world – are there any attempts? With what types of results?

R: To the best of our knowledge, although the Dixon paper has been cited more than 200 times and the scale has been used several times, there are no attempts to validation of OSE outside the Western world. The papers we found that utilized the scale do not mention an adaptation/validation. They declare they used the original version, translated, but not validated. E.g. Xinlei, C., & Yiyang, L. (2022). Online learning engagement in international collaborative nursing students: A questionnaire study. Unpublished thesis. Nursing department, medicine and health college. Lishui University, China.

Data on the reliability of this version have now been reported in the introduction (i.e., lines 168-172)

-        Why can the Italian situation specific from the aspect of engagement? The description of it is a little bit superficial for me. Similar to the demonstration of the circumstances, possibilities and acceptance of online teaching.

R: Thank you for raising this point. We did our best to make it more compelling by providing evidence from the literature that endorses the need for context-relevant instruments for the investigation of online students’ engagement. We addressed this issue both in both in the introduction (i.e., lines 201-211) and discussion section (i.e., 1409-1415).

-        I think the explanation of the selection of UWES can be very useful – why this scale was chosen? Authors should add some information about this scale in the introduction too.

R: The two scales are theoretically related and have some conceptual overlap. Furthermore, to the best of our knowledge, the UWES-S is the only reliable measure of university students’ engagement that has an Italian adaptation and validation. We have added information about this scale within the Introduction (i.e., lines 580-591)

-        The earlier empirical results of OSE scale can be more detailed.

R: We have now added details about empirical results of OSE scale in the Introduction (i.e., lines 103-113).

-        If the UWES was used in Italy earlier, authors should describe these results. I can see in the 25th element of the reference list that this is a validated method in Italy but we have to know other details and results (if there are).

R: We have added information about validation of the scale in Italy and results (i.e., lines 580-591). Furthermore, following comments from reviewer 2 we have also computed a computed reliability index and average variance extraction for the dimensions of the UWES in our sample. These indices showed that the scale was reliable.

-        Is there any example for the co-usage of these too scale? Why did the authors choose?

R: to the best of our knowledge, there is no example for the co-usage of the two scales in the literature. However, we choose this scale as the UWES-S is the only reliable measure of university students’ engagement that has an Italian adaptation and validation. We have added information about this scale within the Introduction (i.e., lines 580-591)

I can see a significant limitation due to the sample and sample method. The engagement during the online courses can be different in the given disciplines and this sample belongs to only two institutions (and we can not see enough detailed the features of these institutions) and they come from only one discipline (Education) and one type of course (Psychology). The features of engagement, the working methods of the teaching can be disparate in the field of the other disciplines – e.g. in Science or Medicine.  I think the authors should empathize this part of the limitation more. The arte of women are very high too (289 from 299).

R: These issues have now been addressed in the limitations section (i.e., lines 1392-1396)

Perhaps the description of the sample can be more detailed (part or full time students), the number of the two subsamples (Reggio Emilia and University of Modena) and the patterns of the other socio-demographic variables (if they are given).

R: In Italian universities, there is no such thing as part or full time students; Modena and Reggio Emilia are two different cities but their university is a single institution; we did not collect other demographics - we addressed this issue in the limitations section (i.e., lines 1397-1400)

Future plans and the brief presentation of later research are recommended at the end of the paper (if the authors have this type of planes). 

R: We agree with reviewer 3 and we have addressed this issue in the limitations and conclusions section (i.e., lines 1408-1554)

typos:

row 69 – Eduopean

row 244 – However (in the middle of the sentence)

row 59 – Universities (in the middle of the sentence)

R: We have checked across the manuscript for all typos and fixed them. Thank you again for all your suggestions.

Round 2

Reviewer 2 Report

Dear Authors

Thank you for your revisions.

This version should be published.